# Artificial intelligence-based point-of-care lung ultrasound for screening COVID-19 pneumoniae: Comparison with CT scans

**Yumi Kuroda[1], Tomohiro Kaneko[2], Hitomi Yoshikawa[1], Saori Uchiyama[2], Yuichi Nagata[1], Yasushi Matsushita[3,4], Makoto Hiki[2,3], Tohru Minamino[2], Kazuhisa Takahashi[1], Hiroyuki Daida[2,5], Nobuyuki Kagiyama[2,5]***

1 Department of Respiratory Medicine, Juntendo University Graduate School of Medicine, Bunkyo-ku, Tokyo, Japan, 2 Department of Cardiovascular Biology and Medicine, Juntendo University Graduate School of Medicine, Bunkyo-ku, Tokyo, Japan, 3 Department of Emergency Medicine, Juntendo University Graduate School of Medicine, Bunkyo-ku, Tokyo, Japan, 4 Department of Internal Medicine and Rheumatology, Juntendo University Graduate School of Medicine, Bunkyo-ku, Tokyo, Japan, 5 Department of Digital Health and Telemedicine R&D, Juntendo University, Bunkyo-ku, Tokyo, Japan

* kgnb_27_hot@yahoo.co.jp

**Data Availability Statement:** The data are not publicly open due to privacy concerns. Sharing the

## Abstract

### Background

Although lung ultrasound has been reported to be a portable, cost-effective, and accurate method to detect pneumonia, it has not been widely used because of the difficulty in its interpretation. Here, we aimed to investigate the effectiveness of a novel artificial intelligence-based automated pneumonia detection method using point-of-care lung ultrasound (AI-POCUS) for the coronavirus disease 2019 (COVID-19).

### Methods

We enrolled consecutive patients admitted with COVID-19 who underwent computed tomography (CT) in August and September 2021. A 12-zone AI-POCUS was performed by a novice observer using a pocket-size device within 24 h of the CT scan. Fifteen control subjects were also scanned. Additionally, the accuracy of the simplified 8-zone scan excluding the dorsal chest, was assessed. More than three B-lines detected in one lung zone were considered zone-level positive, and the presence of positive AI-POCUS in any lung zone was considered patient-level positive. The sample size calculation was not performed given the retrospective all-comer nature of the study.

### Results

A total of 577 lung zones from 56 subjects (59.4 ± 14.8 years, 23% female) were evaluated using AI-POCUS. The mean number of days from disease onset was 9, and 14% of patients were under mechanical ventilation. The CT-validated pneumonia was seen in 71.4% of patients at total 577 lung zones (53.3%). The 12-zone AI-POCUS for detecting CT-validated pneumonia in the patient-level showed the accuracy of 94.5% (85.1%– 98.1%), sensitivity of 92.3% (79.7%– 97.3%), specificity of 100% (80.6%– 100%), positive predictive value of

present data with researchers/companies outside the author's research group is not approved by the IRB at Juntendo University, and IRB application and approval upon an appropriate request are necessary before sharing the data (contact information: Nobuyuki Kagiyama; kgnb_27_hot@yahoo.co.jp and Data Coordinator of Department of Cardiovascular Biology and Medicine; junkanki@juntendo.ac.jp).

**Funding:** Kagiyama and Daida are affiliated with a department funded by Philips Japan, Asahi KASEI Corporation, Inter Reha Co., Ltd, and Toho Holdings Co., Ltd., based on collaborative research agreements. Other authors have no conflict of interest to declare. This does not alter our adherence to PLOS ONE policies on sharing data and materials. This work was partially supported by the Japan Society for Promotion of Science KAKENHI (by the Japanese government), with a Grant Number 21K18086. There was no additional external funding received for this study.

**Competing interests:** Kagiyama and Daida are affiliated with a department funded by Philips Japan, Asahi KASEI Corporation, Inter Reha Co., Ltd, and Toho Holdings Co., Ltd., based on collaborative research agreements. Other authors have no conflict of interest to declare. This does not alter our adherence to PLOS ONE policies on sharing data and materials.

**Abbreviations:** COVID-19, coronavirus disease 2019; POCUS, point-of-care ultrasound; AI-POCUS, artificial intelligence-based point-of-care lung ultrasound; CT, computed tomography.

95.0% (89.6% - 97.7%), and Kappa of 0.33 (0.27–0.40). When simplified with 8-zone scan, the accuracy, sensitivity, and sensitivity were 83.9% (72.2%– 91.3%), 77.5% (62.5%–87.7%), and 100% (80.6%– 100%), respectively. The zone-level accuracy, sensitivity, and specificity of AI-POCUS were 65.3% (61.4%– 69.1%), 37.2% (32.0%– 42.7%), and 97.8% (95.2%– 99.0%), respectively.

## Interpretation

AI-POCUS using the novel pocket-size ultrasound system showed excellent agreement with CT-validated COVID-19 pneumonia, even when used by a novice observer.

## Introduction

The novel coronavirus disease 2019 (COVID-19) emerged in China in 2019, and has since spread rapidly worldwide [1]. This highly infectious disease, which often causes fatal pneumonia, has caused 6 million deaths as of March 2022 [2–4]. This rapid increase in the number of patients with COVID-19 has strained medical resources in many countries [5, 6]. In such situations, detailed screening in medical facilities for all patients is not realistic; as such, there is an increasing demand for portable, easy, and cost-effective screening tools.

Point-of-care lung ultrasound (POCUS) is a simple diagnostic test with high diagnostic accuracy for lung disease [7, 8]. A previous systematic review reported that the presence of multiple B-lines could identify the presence of pneumonia with a sensitivity of 90.4%, and a specificity of 88.4% [9]. Although this examination has the potential to serve as a useful screening tool in COVID-19 patients, lung POCUS is not sufficiently widespread, mainly because of technical difficulties in image interpretation.

Artificial intelligence, especially machine learning including deep learning, is an emerging computer technology that automatically recognizes images with excellent accuracy [10–12]. This technology may have the potential to help interpret medical images and enable such examinations, even by novice observers. In this study, we investigated the effectiveness of a novel artificial intelligence-based automated B-line counting system for detecting COVID-19 pneumonia using a pocket-size ultrasound device [13].

## Methods

### Patient enrollment

We enrolled consecutive patients with COVID-19 who were admitted to our hospital and underwent computed tomography (CT) between August and September 2021. Lung POCUS using a pocket-size ultrasound device (Lumify; Philips Ultrasound, Inc., Bothell, WA) with a novel machine learning-based application (guided B-line quantification; Philips Ultrasound, Inc., Bothell, WA) was performed in all patients within 24 h of the CT scan (artificial intelligence-based point-of-care lung ultrasound; AI-POCUS). This machine learning program was developed by the company and is now commercially available. Detailed development process (training, validation, model architecture, etc.) is not publicly open. Patients aged <20 years and pregnant women were excluded. We further performed AI-POCUS in 15 consecutive subjects without lung diseases who underwent chest CT for the purpose of medical checkup, as controls.

## AI-POCUS and CT protocols

AI-POCUS was performed in a 12-zone manner: eight zones for the ventral and four zones for the dorsal chest wall using a 1.0–4.0 Hz sector probe (S4-1; Philips Ultrasound, Inc., Bothell, WA), with a depth setting of 6–12 cm [8, 14, 15]. As shown in Fig 1, the ventral chest wall was segmented by the midline and anterior axillary lines, and further divided into the upper and lower zones. The dorsal chest wall was divided into the left-right and upper-lower zones. For those who were too sick to change their positions (such as intubated patients), only ventral eight zones were scanned [15]. The AI-POCUS movies were automatically analyzed using a real-time automated B-line counting application. All AI-POCUS scans were performed by a blinded novice observer who had experienced fewer than 10 cases of lung POCUS before the start of the study. This machine-learning-based application automatically counts the number of B-lines in the movie, and records the maximum number of B-lines. Three or more B-lines in each zone were considered abnormal (Fig 2).

Non-contrast chest CT scans were acquired using commercially available equipment (Aquilion ONE, Canon, Japan) at inspiration, and reconstructed as axial images with a 5-mm slice thickness, 5-mm interval, and 120 peak kilovoltage. A board-certified pulmonologist, who was blinded to patient characteristics and AI-POCUS findings, read the CT and diagnosed the presence of pneumonia in each of the 12 zones of the lung that corresponded with the lung POCUS segmentation.

Zone-level concordance was assessed between AI-POCUS and CT results for each zone. In contrast, patient-level AI-POCUS results were defined as negative when the patient had no sign of pneumonia in any of the acquired zones, and patient-level concordance was assessed for each patient.

## Statistical analyses

Data are presented as the mean ± standard deviation or median [$1^{st}$ and $3^{rd}$ quartile] for continuous variables, and as the frequency (%) for categorical variables. Differences in patient characteristics between groups were tested using the Welch t-test and Mann-Whitney test for normally and non-normally distributed variables, respectively, and Fisher's exact test for categorical variables. The primary endpoint of the study was patient-level accuracy, sensitivity, and specificity of AI-POCUS in comparison with CT-validated pneumonia. Wilson's method was

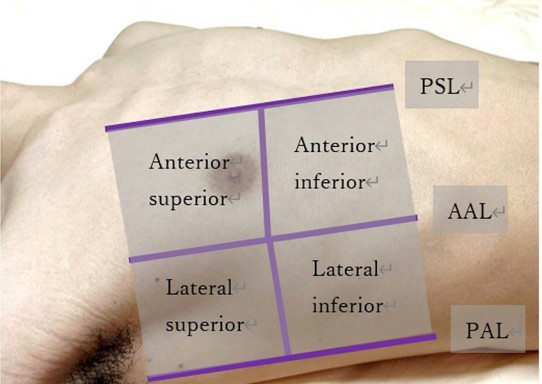 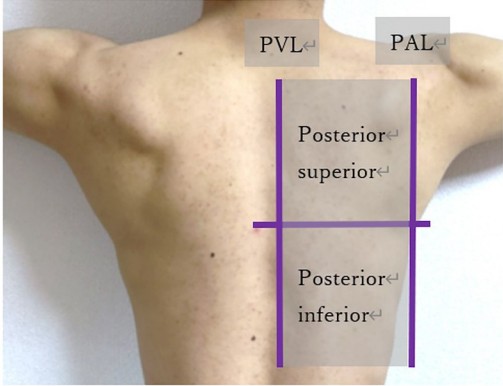

**Fig 1. Lung segmentations in 12-zone lung POCUS.** The chest was divided into eight zones for the ventral wall and four zones for the dorsal chest wall, divided by the posterior axillary line (PAL). The parasternal line (PSL) and paravertebral line (PVL) to two hemithorax. The ventral chest wall was segmented by the midline and anterior axillary lines (AAL) and further divided into upper and lower zones. The dorsal chest wall was divided into left-right and upper-lower zones at the height of the tip of the scapula.

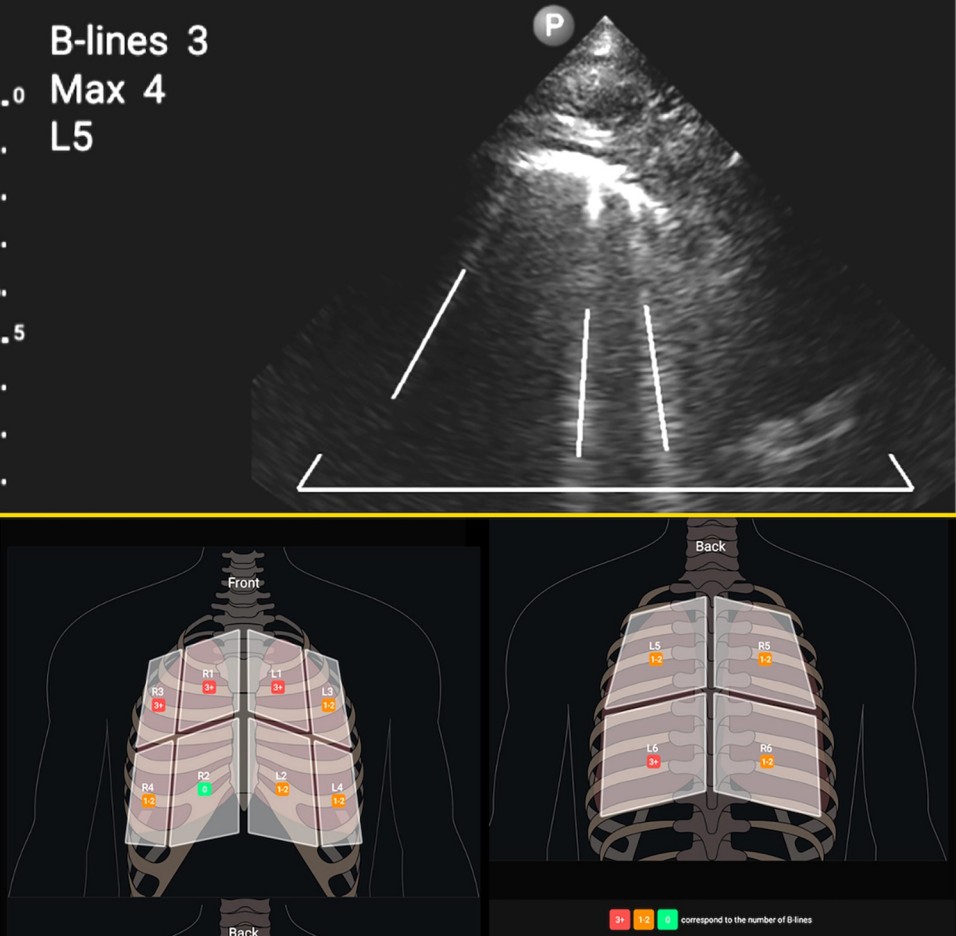

**Fig 2. Ultrasound B-lines.** B-lines were automatically analyzed using the real-time deep learning-based application, which counted the number of B-lines (indicated by the white lines in the upper panel) in the movie, and recorded the maximum number of B-lines. Three or more B-lines in each zone were considered abnormal (red zone in lower panel). Patients with one or more abnormal zones were considered positive for AI-POCUS.

used to calculate the 95% confidence interval for each metric [16]. All statistical analyses were performed with R version 4.0.2 (The R Foundation for Statistical Computing, Vienna, Austria). A two-tailed p value of $<0.05$ indicated statistical significance.

## Ethics

The study protocol complied with the Declaration of Helsinki and was approved by the institutional review board (IRB) at Juntendo University Hospital (#E21-0197). Written informed consent was waived due to its purely observational nature by the IRB based on the "Ethical Guidelines for Medical and Health Research Involving Human Subjects" issued by Japanese Ministry of Health, Labor and Welfare.

## Results

### Study population

During the study period, 45 patients with COVID-19 were admitted to the hospital. After excluding 4 patients who did not undergo lung POCUS due to admission on weekends, 41

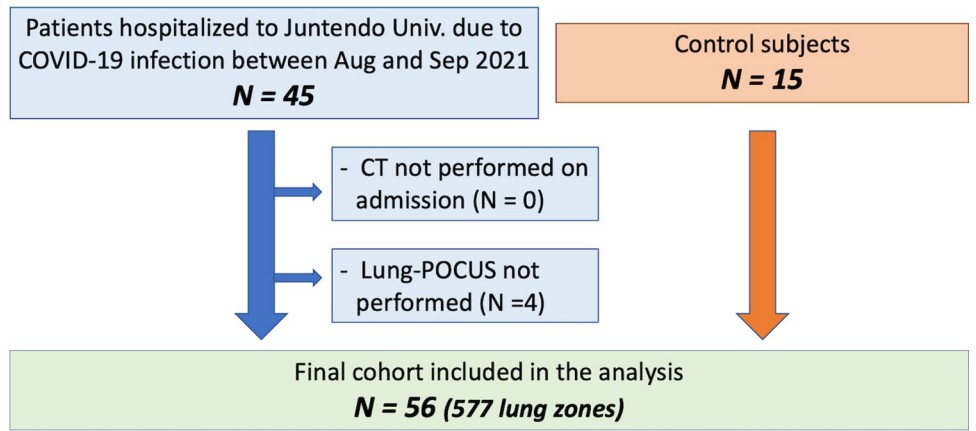

**Fig 3. Patient inclusion chart.** Only 4 patients (8.9%) of cases were excluded and 56 patients (41 cases and 15 controls) were included in the final analysis.

patients were enrolled in the study. Including 15 control subjects, a total of 56 subjects (41 patients and 15 controls) were included in the study (Fig 3). The baseline patient characteristics are summarized in Table 1. Overall, the mean age was 59.4 ± 14.8 years, 23% were female, and the mean body mass index was 25.3 ± 4.7 kg/m$^2$. The patients were significantly younger than the controls. The median time from the onset to the CT scan was 9 [5 – 12] days, and 20% of the patients were maintained on mechanical ventilation at the time of CT and AI-POCUS.

## AI-POCUS and CT availability

CT evaluation was successful for all 12 zones for all subjects (492 zones for cases and 180 for controls), whereas lung POCUS was not available for the dorsal chest zones in 15 patients who were unable to change their positions. One patient was placed in the prone position, and only the dorsal zones were scanned. Clear images were not available in the left lower inside zone in the ventral chest of 26 patients because the heart was close to the chest wall, and the lung was therefore not visible from this position. Finally, lung POCUS movies of 397 zones for patients and 180 zones for controls were analyzed.

## Concordance of pneumonia identification

A CT scan identified pneumonia in 395 zones (58.7%) of 40 subjects, whereas no pneumonia was detected in the other 16 subjects. Pneumonia was present almost evenly in each zone, with a slightly higher prevalence in the dorsal zones (Fig 4).

Panel A in Fig 5 demonstrates the patient-level concordance between the AI-POCUS and CT-validated pneumonia. One subject was excluded from this analysis because AI-POCUS

**Table 1. Patient characteristic.**

|  | Overall (N = 56) | Case (N = 41) | Control (N = 15) | p value |
|---|---|---|---|---|
| Age, years old | 59.4 ± 14.8 | 56.3 ± 14.6 | 67.6 ± 12.5 | 0.008 |
| Female | 13 (23%) | 8 (20%) | 5 (33%) | 0.30 |
| Body mass index, kg/m$^2$ | 25.3 ± 4.7 | 25.6 ± 5.1 | 24.7 ± 3.5 | 0.50 |
| Days from onset |  | 9.0 [5.0–12.0] |  |  |
| Mechanical Ventilation |  | 8 (20%) |  |  |

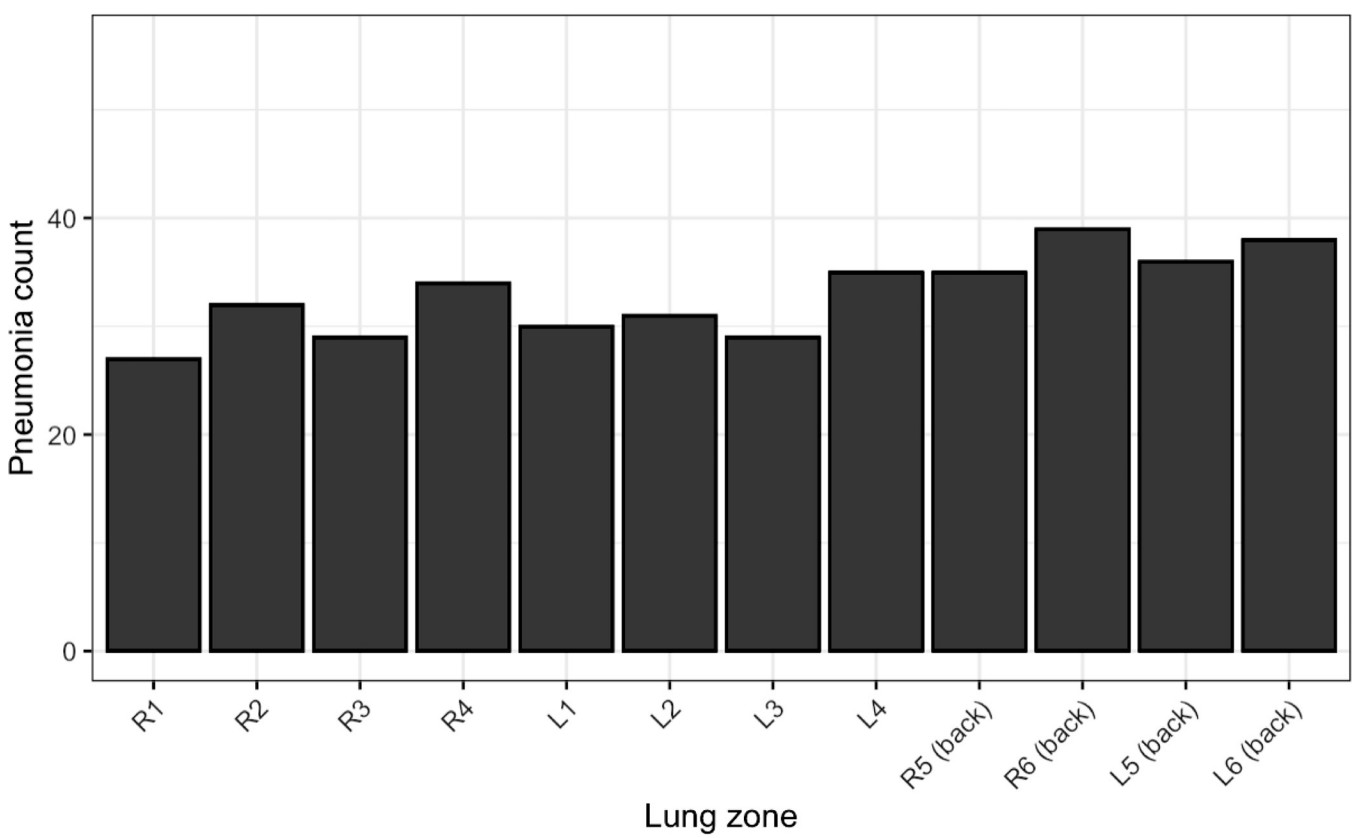

**Fig 4. Distribution of pneumonia.** The X-axis represents the zone of the lung (R1, right anterior superior zone; R2, right anterior inferior zone; R3, right lateral superior zone; R4, right lateral inferior zone; R5, right superior dorsal zone; R6, right inferior dorsal zone; Ls are left zones corresponding to Rs), and the Y-axis represents the number of patients with pneumonia in each zone.

was not possible for the dorsal zones, although he only had pneumonia there. AI-POCUS was able to diagnose the presence or absence of pneumonia in 52 of 55 subjects, while pneumonia was overlooked in three patients. The overall accuracy, sensitivity, specificity, PPV, NPT, and Kappa of the AI-POCUS were 94.5% (85.1%– 98.1%), 92.3% (79.7%– 97.3%), 100% (80.6%– 100%), 100% (90.4%– 100%), 84.2% (62.4–94.5%), and 0.89 (0.70–0.98), respectively.

The screening performance of AI-POCUS using only the ventral eight zones was evaluated as a simpler approach that can be used even for those who cannot change their position. With only 8-zone AI-POCUS, pneumonia was overlooked in 9 patients, whereas 31 of 40 patients with pneumonia were still classified as positive (Fig 5, Panel B). The accuracy, sensitivity, NPV, and Kappa decreased to 83.9% (72.2%– 91.3%), 77.5% (62.5%– 87.7%), 64.0% (44.5–79.8%), and 0.68 (0.43–0.85), respectively; however, the specificity and PPV remained at 100% for both (80.6%– 100% and 89.0%– 100%).

In the zone-level analysis, AI-POCUS was correct in 377 of 577 zones (Fig 5, Panel C). The zone-level accuracy, sensitivity, specificity, PPV, NPV, and Kappa of the AI-POCUS were 65.3% (61.4%– 69.1%), 37.2% (32.0%– 42.7%), 97.8% (95.2%– 99.0%), 95.5% (89.6%– 97.7%), 57.5% (52.9%– 61.9%), and 0.31 (0.23–0.38), respectively.

Finally, we assumed that the prevalence of pneumonia in COVID-19 positive patients is 5% based on the public data from the Japanese Ministry of Health, Labour and Welfare and calculated the estimated PPV and NPV in the real-world. Given the sensitivity of 92.3% and

## A. Patient-level accuracy (12-zone)

|  | CT negative | CT positive |
|---|---|---|
| POCUS negative | 16 | 3 |
| POCUS positive | 0 | 36 |

## C. Zone-level accuracy

|  | CT negative | CT positive |
|---|---|---|
| POCUS negative | 262 | 194 |
| POCUS positive | 6 | 115 |
| POCUS Infeasible | 9 | 86 |

## B. Patient-level accuracy (8-zone)

|  | CT negative | CT positive |
|---|---|---|
| POCUS negative | 16 | 9 |
| POCUS positive | 0 | 31 |

**Fig 5. Accuracy of AI-POCUS versus CT.** Each panel is a confusion matrix showing the concordance between the AI-POCUS results and CT-validated pneumonia diagnosis.

specificity of 100%, and 5% of prior probability, PPV was 100% (99.9%– 100%) and NPV was 99.6% (99.6%– 99.6%).

## Discussion

The main findings of our study were as follows: (1) AI-POCUS had excellent performance at detecting patients with COVID-19 pneumonia (sensitivity 92.3% and specificity 100%), and (2) the zone-level sensitivity was moderate (37.3%), although specificity was very high (97.8%). Even with moderate sensitivity in each zone, pneumonia in COVID-19 usually spreads to multiple lung zones, leading to a high sensitivity for each patient. Notably, these excellent results were achieved with AI-POCUS by a novice observer who had minimal experience in lung-POCUS.

With the recent miniaturization of ultrasound devices and advancements in image quality, lung-POCUS is becoming a popular examination, especially in intensive care [8, 17]. Although the lung itself, which is filled with air, cannot usually be observed by ultrasound beams, the noise and artifacts generated by ultrasound beams provide useful clinical information in lung-POCUS. Ultrasound B-lines are one of the most useful artifacts that present with pulmonary

congestion, either by pneumonia or cardiogenic pulmonary edema. These reverberation artifacts originate at the pleura, reflecting an air-fluid mixture, which occurs when the subpleural interlobular septa surrounded by subpleural air-filled alveoli become edematous. Previous studies have reported that three or more B-lines visible in a single ultrasound plane are fully sensitive and specific to demonstrate subpleural thickened interlobular septa and/or ground-glass areas with a CT scan as a reference [18].

Diagnosis of pneumonia using lung-POCUS techniques, including the B-line, is expected to be an effective tool in the ongoing COVID-19 pandemic. Reports have already shown that B-lines are not only sensitive, but also associated with disease severity, future deterioration, and treatment effects in COVID-19 [15, 19–23]. However, chest radiography and CT remain the leading examinations used in the management of COVID-19 worldwide, and lung-POCUS has not been sufficiently used. Technical difficulties and problems in interpreting images are major concerns when using lung-POCUS; the number of experts is not large enough to teach and supervise the use of lung-POCUS by clinicians, including general practitioners, although this technique is relatively new and may be difficult for novice observers.

In the present study, we demonstrated that with AI technology, lung-POCUS can be effective to diagnose pneumonia, with excellent accuracy, even by a novice observer. Recent advancements in AI technology, more specifically machine learning, have enabled automated image recognition with similar or even higher accuracy compared to expert clinicians, and the application of such technologies to POCUS has been enthusiastically studied [10, 11, 24, 25]. The automated B-line counting application that we used in this study was also developed using machine learning technology, and showed excellent agreement with expert readings [26, 27].

The moderate zone-level accuracy observed herein may be due to the limited ability to detect slight pneumonia in the central area of the lung far from the body surface. Such slight pneumonia in a deep area of the lung may not affect the pleura, and would therefore not generate a B line. However, pneumonia caused by COVID-19 usually spreads to multiple regions of the lungs, appearing in the form of acute respiratory distress syndrome [23]. Therefore, diffuse pneumonia caused by COVID-19 is a suitable target for AI-POCUS.

COVID-19 is expected to persist for some time. AI-POCUS allows non-specialist general practitioners to diagnose pneumonia, a problem in viral infections, in an access-free manner, which may be useful in the treatment of pneumonia during a pandemic, and is expected to become more widespread.

## Limitations

This study has some limitations. First, this was a single-center study conducted in a university hospital, and included a small number of cases. Larger studies involving various settings are necessary before the current results can be widely implemented. Second, we did not check the accuracy of AI-POCUS compared with traditional lung POCUS by an expert. However, a previous study showed that AI-POCUS has over 90% sensitivity and specificity for detecting expert-annotated B-lines [26]. Next, we studied only hospitalized patients who were considered sicker than those who were cared for at home or at other facilities. The performance of the present application should be tested in less intensively sick patients to uncover it's applicability for screening purposes.

## Conclusions

AI-POCUS showed excellent performance in diagnosing patients with CT-validated COVID-19 pneumonia, even by a novice observer.

## Author Contributions

**Conceptualization:** Saori Uchiyama, Nobuyuki Kagiyama.

**Data curation:** Yumi Kuroda, Tomohiro Kaneko, Hitomi Yoshikawa, Saori Uchiyama, Yuichi Nagata, Yasushi Matsushita, Makoto Hiki, Nobuyuki Kagiyama.

**Investigation:** Tomohiro Kaneko, Nobuyuki Kagiyama.

**Methodology:** Nobuyuki Kagiyama.

**Resources:** Tohru Minamino, Kazuhisa Takahashi, Hiroyuki Daida.

**Supervision:** Nobuyuki Kagiyama.

**Validation:** Nobuyuki Kagiyama.

**Visualization:** Nobuyuki Kagiyama.

**Writing – original draft:** Yumi Kuroda, Tomohiro Kaneko, Hitomi Yoshikawa, Saori Uchiyama, Nobuyuki Kagiyama.

**Writing – review & editing:** Yumi Kuroda, Tomohiro Kaneko, Hitomi Yoshikawa, Saori Uchiyama, Yuichi Nagata, Yasushi Matsushita, Makoto Hiki, Tohru Minamino, Kazuhisa Takahashi, Hiroyuki Daida, Nobuyuki Kagiyama.

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
