## [Decision Letter · Decision Letter 0]

26 Sep 2022

PONE-D-22-21871Artificial Intelligence-based Point-of-care Lung Ultrasound for Screening COVID-19 Pneumoniae: Comparison with CT ScansPLOS ONE

Dear Dr. Kagiyama,

Thank you for submitting your manuscript to PLOS ONE. After careful consideration, we feel that it has merit but does not fully meet PLOS ONE’s publication criteria as it currently stands. Therefore, we invite you to submit a revised version of the manuscript that addresses the points raised during the review process.

We look forward to receiving your revised manuscript.

Kind regards,

Juan Antonio Valera-Calero

Academic Editor

PLOS ONE

Journal Requirements:

"This work was partially supported by JSPS KAKENHI, with a Grant Number 21K18086."

Please expand the acronym “JSPS KAKENHI” (as indicated in your financial disclosure) so that it states the name of your funders in full.

"Kagiyama and Daida are affiliated with a department funded by Philips Japan, Asahi KASEI Corporation, Inter Reha Co., Ltd, and Toho Holdings Co., Ltd., based on collaborative research agreements. Other authors have no conflict of interest to declare."

Additional Editor Comments:

Please address the comments provided by the reviewers.

Reviewers' comments:

Reviewer's Responses to Questions

**Comments to the Author**

1. Is the manuscript technically sound, and do the data support the conclusions?

Reviewer #1: Partly

Reviewer #2: Yes

2. Has the statistical analysis been performed appropriately and rigorously? 

Reviewer #1: No

Reviewer #2: I Don't Know

3. Have the authors made all data underlying the findings in their manuscript fully available?

Reviewer #1: No

Reviewer #2: Yes

4. Is the manuscript presented in an intelligible fashion and written in standard English?

Reviewer #1: Yes

Reviewer #2: Yes

5. Review Comments to the Author

Reviewer #1: Abstract:

The CI 95% of the performance indices are missing. Moreover, the PPV, NPV, and Kappa must be added as important performance indices to the abstract and paper.

The percentage of CT-validated pneumonia must be added to the subject (65) and frame data (577 lung zones).

If random sampling and proper sample size estimation are not used, they must be mentioned as important issues that avoid the generalization to the entire population in the abstract and discussion.

Material and Methods:

The flowchart of the experimental design (including the inclusion and exclusion criteria, etc.) must be added to the paper.

Statistical analysis:

The CI 95% must be added to the entire indices based on the TRIPOD/STARD guideline. The Kappa (agreement rate) and PPV (the probability of having pneumonia if the test result is positive) must be added.

The authors mentioned that the AI method's training, validation, and further details are unavailable. A scientific work must provide the required information to repeat the work. For example, it is unclear which validation framework (hold-out, cross-validation, etc.) was used and whether the results are based on the cross-validated confusion matrix. Moreover, the percentage of the training, validation, and test sets must be provided.

The AI method must be briefly described in the paper, and providing the citation is insufficient.

Results:

For the entire indices (plus PPV, NPV, and Kappa), the CI 95% must be provided, showing the precision of the indices.

The objective comparison with the state-of-the-art using proper statistical methods is missing.

A major issue is how the proposed method performs in practice. The authors must use the sensitivity and specificity of the proposed system and the prevalence of pneumonia in the population (based on the literature) and use the Bayesian theorem to predict the PPV and NPV of the system in the population.

Section 2 of the following paper could be easily used by the authors for this estimation:

https://www.ncbi.nlm.nih.gov/pmc/articles/PMC8170576/

Discussion:

And finally, is the proposed diagnosis system clinically reliable?

Reviewer #2: Dear author,

The strength of the paper is well structured, the description of the related work is well done and that results are compared to results of the similar research. At all the work is of certain significance. The paper provides a very powerful message about the "Artificial Intelligence-based Point-of-care Lung Ultrasound for Screening COVID-19 Pneumoniae: Comparison with CT Scans".

6. PLOS authors have the option to publish the peer review history of their article (what does this mean?). If published, this will include your full peer review and any attached files.

Reviewer #1: **Yes: **Hamid Reza Marateb

Reviewer #2: No

---

## [Author Response · Author response to Decision Letter 0]

16 Dec 2022

Review Comments to the Author

Reviewer #1: 

Abstract:

The CI 95% of the performance indices are missing. Moreover, the PPV, NPV, and Kappa must be added as important performance indices to the abstract and paper.

Response: Thank you for the valuable suggestion. We calculated the confidence intervals using the Wilson’s method (1), which is recommended by several papers (2, 3), as shown below. We were not able to include all metrics for all examinations (i.e. for full-zone patient-level, 8-zone patient-level, and zone-level) in the abstract, so PPV and Kappa for the secondary outcomes were included only in the manuscript text. 

1 E.B. Wilson. Probable inference, the law of succession, and statistical inference. J Am Stat Assoc 1927;22:209–212.

2 A. Agresti and B.A. Coull. Approximate is better than "exact" for interval estimation of binomial proportions. Am Statistician, 1998;52:119-126.

3 L.D. Brown, T.T. Cai and A. Dasgupta Interval estimation for a binomial proportion Statistical Science, 2001;16(2):101-133

[Original in Abstract]

The patient-level accuracy of 12-zone AI-POCUS for detecting CT-validated pneumonia was 94.5%, sensitivity was 92.3%, and specificity was 100%. When simplified with 8-zone scan, the accuracy, sensitivity, and sensitivity were 83.9%, 77.5%, and 100%, respectively. The zone-level accuracy, sensitivity, and specificity of AI-POCUS were 65.3 %, 37.2%, and 97.8 %, respectively.

[Revised]

The 12-zone AI-POCUS for detecting CT-validated pneumonia in the patient-level showed the accuracy of 94.5% (85.1% – 98.1%), sensitivity of 92.3% (79.7% – 97.3%), specificity of 100% (80.6% – 100%), positive predictive value of 95.0% (89.6% - 97.7%), and Kappa of 0.33 (0.27 – 0.40). When simplified with 8-zone scan, the accuracy, sensitivity, and sensitivity were 83.9% (72.2% – 91.3%), 77.5% (62.5% – 87.7%), and 100% (80.6% – 100%), respectively. The zone-level accuracy, sensitivity, and specificity of AI-POCUS were 65.3% (61.4% – 69.1%), 37.2% (32.0% – 42.7%), and 97.8 % (95.2% – 99.0%), respectively.

The percentage of CT-validated pneumonia must be added to the subject (65) and frame data (577 lung zones).

Response: We included them in the manuscript.

[Added in Abstract]

The CT-validated pneumonia was seen in 71.4% of patients at total 577 lung zones (53.3%).

If random sampling and proper sample size estimation are not used, they must be mentioned as important issues that avoid the generalization to the entire population in the abstract and discussion.

Response: We added the following limitation in the abstract as well as discussion.

[Added in Abstract]

The sample size calculation was not performed given the retrospective all-comer nature of the study.

Material and Methods:

The flowchart of the experimental design (including the inclusion and exclusion criteria, etc.) must be added to the paper.

Response: We added the following flow chart as new Figure 1.

[Original in Results]

A total of 56 subjects (41 patients and 15 controls) were enrolled in the study.

[Added in Results]

During the study period, 45 patients with COVID-19 were admitted to the hospital. After excluding 4 patients who did not undergo lung POCUS due to admission on weekends, 41 patients were enrolled in the study. Including 15 control subjects, a total of 56 subjects (41 patients and 15 controls) were included in the study (Figure 1).

Statistical analysis:

The CI 95% must be added to the entire indices based on the TRIPOD/STARD guideline. The Kappa (agreement rate) and PPV (the probability of having pneumonia if the test result is positive) must be added.

Response: Throughout the manuscript, we added the 95% confidence intervals for the performance metrics and PPV and Kappa values. The changes in the results are shown below. The additional statistical analysis was explained as shown below.

[Added in the statistical analysis]

Wilson’s method was used to calculate the 95% confidence interval for each metric (1).

1 E.B. Wilson. Probable inference, the law of succession, and statistical inference. J Am Stat Assoc 1927;22:209–212.

[Original (overall patient-level analysis)]

The overall accuracy, sensitivity, and specificity of the AI-POCUS were 94.5%, 92.3%, and 100%, respectively.

[Revised]

The overall accuracy, sensitivity, specificity, PPV, NPT, and Kappa of the AI-POCUS were 94.5% (85.1% – 98.1%), 92.3% (79.7% – 97.3%), 100% (80.6% – 100%), 100% (90.4% – 100%), 84.2% (62.4 – 94.5%), and 0.89 (0.70 – 0.98), respectively. 

[Original (patient-level analysis using only 8 zones)]

The accuracy and sensitivity decreased to 83.9% and 77.5%, respectively; however, the specificity remained at 100%.

[Revised]

The accuracy, sensitivity, NPV, and Kappa decreased to 83.9% (72.2% – 91.3%), 77.5% (62.5% – 87.7%), 64.0% (44.5 – 79.8%), and 0.68 (0.43 – 0.85), respectively; however, the specificity and PPV remained at 100% for both (80.6% – 100% and 89.0% – 100%).

[Original (zone-level)]

The zone-level accuracy, sensitivity, and specificity of the AI-POCUS were 65.3%, 37.2%, and 97.8%, respectively.

[Revised]

The zone-level accuracy, sensitivity, specificity, PPV, NPV, and Kappa of the AI-POCUS were 65.3% (61.4% – 69.1%), 37.2% (32.0% – 42.7%), 97.8% (95.2% – 99.0%), 95.5% (89.6% – 97.7%), 57.5% (52.9% – 61.9%), and 0.31 (0.23 – 0.38), respectively.

The authors mentioned that the AI method's training, validation, and further details are unavailable. A scientific work must provide the required information to repeat the work. For example, it is unclear which validation framework (hold-out, cross-validation, etc.) was used and whether the results are based on the cross-validated confusion matrix. Moreover, the percentage of the training, validation, and test sets must be provided.

The AI method must be briefly described in the paper, and providing the citation is insufficient.

Response: Thank you for letting us clarify this important point. Although we used an AI application in this study, this development process was performed by the company (Philips) and is not publicly open. In this study, we applied this already-developed commercially available software to our population that is completely different from what the company used to develop the model. In this term, this study can be considered as an external or real-world validation of the model, not the development of the model. As a validation study, we believe that the pipeline and study flow of our study are in line with other studies (e.g., references 1-3) and are in a scientific manner.

1. C. Dung-Hung, et al. Crit Care 2022;26(1):215

2. IZ Attia, at al. Int J Cardiol 2021;329:130-135.

3. N, Kagiyama, et al. Echocardiography. 2016;33(5):756-63

Results:

For the entire indices (plus PPV, NPV, and Kappa), the CI 95% must be provided, showing the precision of the indices.

Response: We added 95% confidence indices and PPV, NPV and Kappa in results as shown above in the response to the previous comment.

The objective comparison with the state-of-the-art using proper statistical methods is missing.

Response: As we explained in the previous response. This is not a study in which we developed a machine-learning model, but we performed an external validation of the machine learning-based application that was already established and fixed. The gold standard was the CT results, and there is no state-of-the-art to be compared with.

A major issue is how the proposed method performs in practice. The authors must use the sensitivity and specificity of the proposed system and the prevalence of pneumonia in the population (based on the literature) and use the Bayesian theorem to predict the PPV and NPV of the system in the population.

Section 2 of the following paper could be easily used by the authors for this estimation:

https://www.ncbi.nlm.nih.gov/pmc/articles/PMC8170576/

Response: We thank the reviewer for this valuable suggestion. The hospitalization rate of COVID-19 in the elderly people in Japan is expected to be 5 – 15% by the Japanese Ministry of Health, Labour and Welfare (https://www.mhlw.go.jp/content/10900000/000964719.pdf). Based on this data, we assumed that the prevalence of pneumonia is 5% in COVID-19 positive patients. Given the following parameters, we calculated the PPV and NPV. 

Given the sensitivity of 92.3% and specificity of 100%, and 5% of prior probability;

PPV: 100% (99.9% – 100%)

NPV: 99.6% (99.6% – 99.6%) 

We added these results to the manuscript.

[Added in Results]

Finally, we assumed that the prevalence of pneumonia in COVID-19 positive patients is 5% based on the public data from the Japanese Ministry of Health, Labour and Welfare and calculated the estimated PPV and NPV in the real-world. Given the sensitivity of 92.3% and specificity of 100%, and 5% of prior probability, PPV was 100% (99.9% – 100%) and NPV was 99.6% (99.6% – 99.6%).

Discussion:

And finally, is the proposed diagnosis system clinically reliable?

Response: Yes. The present study included consecutive patients with the exclusion of less than 10%. We believe that the population represents the real-world COVID-19 population, and the present results are generalizable in patients hospitalized with COVID-19. The good patient-level accuracy is clinically meaningful. However, as we stated in the manuscript as below, there are limitations and further studies are warranted.

[Original in the limitation]

First, this was a single-center study conducted in a university hospital, and included a small number of cases. Larger studies involving various settings are necessary before the current results can be widely implemented. ~~ Next, we studied only hospitalized patients who were considered sicker than those who were cared for at home or at other facilities. The performance of the present application should be tested in less intensively sick patients to uncover it’s applicability for screening purposes.

Reviewer #2: 

The strength of the paper is well structured, the description of the related work is well done and that results are compared to results of the similar research. At all the work is of certain significance. The paper provides a very powerful message about the "Artificial Intelligence-based Point-of-care Lung Ultrasound for Screening COVID-19 Pneumoniae: Comparison with CT Scans".

Response: Thank you for your very encouraging comment. We are very happy that you stated that our paper is of certain significance and provides a very powerful message. We’d like to express our appreciation.

---

## [Decision Letter · Decision Letter 1]

16 Jan 2023

Artificial Intelligence-based Point-of-care Lung Ultrasound for Screening COVID-19 Pneumoniae: Comparison with CT Scans

PONE-D-22-21871R1

Dear Dr. Kagiyama,

We’re pleased to inform you that your manuscript has been judged scientifically suitable for publication and will be formally accepted for publication once it meets all outstanding technical requirements.

Kind regards,

Juan Antonio Valera-Calero

Academic Editor

PLOS ONE

Additional Editor Comments (optional):

Reviewers' comments:

Reviewer's Responses to Questions

**Comments to the Author**

1. If the authors have adequately addressed your comments raised in a previous round of review and you feel that this manuscript is now acceptable for publication, you may indicate that here to bypass the “Comments to the Author” section, enter your conflict of interest statement in the “Confidential to Editor” section, and submit your "Accept" recommendation.

Reviewer #1: All comments have been addressed

2. Is the manuscript technically sound, and do the data support the conclusions?

Reviewer #1: Yes

3. Has the statistical analysis been performed appropriately and rigorously? 

Reviewer #1: Yes

4. Have the authors made all data underlying the findings in their manuscript fully available?

Reviewer #1: No

5. Is the manuscript presented in an intelligible fashion and written in standard English?

Reviewer #1: Yes

6. Review Comments to the Author

Reviewer #1: The authors responded to the entire issues addressed by the reviewer. The paper is now suitable for publication.

7. PLOS authors have the option to publish the peer review history of their article (what does this mean?). If published, this will include your full peer review and any attached files.

Reviewer #1: **Yes: **Hamid Reza Marateb

---

## [Editor Report · Acceptance letter]

30 Jan 2023

PONE-D-22-21871R1 

Artificial Intelligence-based Point-of-care Lung Ultrasound for Screening COVID-19 Pneumoniae: Comparison with CT Scans 

Dear Dr. Kagiyama:

I'm pleased to inform you that your manuscript has been deemed suitable for publication in PLOS ONE. Congratulations! Your manuscript is now with our production department. 

Kind regards, 

on behalf of

Dr. Juan Antonio Valera-Calero 

Academic Editor

PLOS ONE